# Agriculture and Green Economy for Environmental Kuznets Curve Adoption in Developing Countries: Insights from Rwanda

**Jean Niyigaba [1,2,\*]**, **Jessica Ya Sun [1]**, **Daiyan Peng [1]** and **Clemence Uwimbabazi [3]**

[1]   School of Economics, Huazhong University of Science and Technology, Wuhan 430074, China; jessicasunya@hust.edu.cn (J.Y.S.); pengdaiyan0880@126.com (D.P.)
[2]   College of Business and Economics, University of Rwanda, Kigali 4285, Rwanda
[3]   Department of Civil, Environmental, and Geomatic Engineering, University of Rwanda, Kigali 4285, Rwanda; uwiclemence6@gmail.com
\*   Correspondence: niyigav@yahoo.fr

**Abstract:** Development and climate change are crucial global concerns with significant contrasts between developed and developing nations. Contrary to several developing countries, Rwanda opted for a green growth policy pathway while struggling with its economic emergence through the alternative green sectors, including agriculture. No research has yet been conducted on the choice's performance on emission sequestration or the country's income, allowing the formulation of strategies accordingly. The environmental Kuznets curve (EKC), mostly adopted by developed countries, is applied for the Rwandese scenario to verify its adoption in developing countries. The within and between effects of the agricultural sector (AGRc) and gross domestic products (GDPc) on $CO_2$ emission ($CO_2$) are examined with an Autoregressive Distributed Lag (ARDL) cointegration and coupling methods in January 2008−December 2018 period. Results confirm the short-run and long-run cointegration relationships of variables, where $CO_2$-GDPc and $CO_2$-AGRc are relatively decoupling and absolute decoupling, respectively. The EKC adoption to $CO_2$-GDPc relationship, and the significant negative causality from GDPc and AGRc to $CO_2$, are confirmed. The performance resulted from the country's environment conservation policies, and Rwanda is a learning example as a developing country. However, the green economy through the agro-economy is at a low level and should be reinforced.

**Keywords:** agro-economy; autoregressive distributed lag; coupling; environmental Kuznets curve

## 1. Introduction

Nowadays, sustainable development and environmental protection are crucial worldwide concerns, where transitions towards sustainable production and consumption are buzzwords [1]. The agriculture sector produces several environmental protection services, including greenhouse gas sequestration. Thus, an agro-economy pathway is a significant green economy promotion, despite several challenges facing the sector, including climate change and its adaptation to technological innovations [2]. A green economy is largely analyzed through the $CO_2$ emissions with the nexus of income growth determinants [3]; the linkage was revealed by the environmental Kuznets curve (EKC) theory supporting an inverted U-shape relation [4]. That theory assumes that the environment tends to worsen as new economic growth occurs until the income average reaches a specific development point. Thus, the EKC is mostly applicable to developed nations, as the ones that are supposed to be at that point. Consequently, several developing nations argue that it is reasonable for them to anticipate

economic growth by environmental degradation as their degradation is minimal compared to the advanced nations [5].

Contrary to several developing countries, after the conception of Rwanda Environment Management Authority (REMA) in 2006 [6], Rwanda prefers the green economy approach, with various environmental conservation policies and specified implementation plans [7,8]. Simultaneously, it strived with emergence through the alternative green economy sectors, including agriculture [9]. The country discouraged environmental degradation activities despite their GDP contribution shares, where their increased by only 1%, while the green economy shares to GDP raised by 4% in 2006–2019 [10]. Meanwhile, the country held stable GDP growth, averaging 7.5% annually in 2000–2018, where the agriculture is a crucial sector offering 30% of the GDP and hiring 70% of the workforce. Moreover, improved farming policies were adopted, and the budget allocated in the sector rose about two times in 2009–2018 [11]. However, its GDP shares decreased from 33% to 28%, the workforce from 88.6% to 69.8% in 2006–2018, and mainly with the unskilled older people [12].

The above contradictions of agriculture performance and green economy initiatives, raised our research concerns to verify the inconsistency causes and significance of those green initiatives on income and green growth. A green economy policy is a low carbon resource-efficient and an alternative model enhancing human welfare by decreasing environmental hazards. It requires an economic transformation to sustainability drivers. This preference is critical for some governments, especially developing countries, having the major challenges of environmental protection while keeping economic stability, as environmental stocks are their main economic providers [13].

The Rwandese green economy and agriculture policies allow scholars to write on this subject. According to Bagstad et al. (2020), Rwanda implemented the ecosystem accounts balancing the income growth with environmental protection [14]. Moreover, Nishimwe et al. (2020), noted that those initiatives mostly elaborated in developed countries allow integrating natural resources into the national accounts system. Thus, the experience could be useful to other countries [15]. Government green and economy monitors also evaluate the policy performance, where the reduction greenhouse gases emission is 2% in 2015–2020 period [16], while 137,500 green jobs for income growth were created [17]. However, Weatherspoon et al. (2019) reveal stunting as a major problem, especially in rural areas, despite the impressive economic growth, thus encouraging the sector's transformations to improve food production [18]. Furthermore, in forecasting, it was discovered that despite the general agriculture prediction increment, the increment rate is low compared to the preceding years due to several inefficient capability issues. Therefore, a transformation plan for investment promotion and opportunity awareness to the competent workforce was suggested [12].

Beside the literature on Rwanda, several authors also wrote broadly on green economy issues, especially the agro-economy subject while their conclusions are mostly converging with the Rwandese scenarios. Focusing on the landscape analysis and conditions that could support the agriculture sector towards a sustainable status, Marcello et al. (2020) denoted that many transformation strategies are required, including private investment attraction, where policymakers should create an adequate financial profitability environment with favorite legislation, transparency, and infrastructure development [19]. However, with a systematic review of academic contributions on a green economy, it was denoted that no convincing empirical studies established that could adequately engage investors in green sectors, suggesting more empirical researches with mixed and refined methodologies [20].

Preceding scholars wrote on the Rwandese agriculture and green economy performance. However, little concern has been given to investigate the green economy policies' performance on greenhouse gas emissions sequestration, especially the agriculture sector performance on emission sequestration, to allow for the reformulation of strategies and policies accordingly. Furthermore, as mentioned by the previous literature, a lack convincing empirical studies, including economic models, also restricts that evaluation. Thus, this research applies the theoretical and empirical approaches to overcome the issues, where an ARDL bounds test of cointegration with coupling hypothesis, and ECM causality models on $CO_2$, AGRc, and GDPc from 2008 to 2018 allow the following aims:

- Identify whether developing countries with green economy policies like Rwanda could adopt the EKC, mostly adopted by advanced counties. Through the relationship classification between $CO_2$-GDPc and $CO_2$-AGRc. As either relative decoupling, absolute decoupling, relative coupling, absolute coupling, or no relationship.
- Explore the GDPc and AGRc impact on $CO_2$ emissions;
- Precise the causality directions through short-run and long-run stability;
- Classify the existing weaknesses, opportunities, and improvement strategies.

The rest of this paper is designed as follows: Section 2 outlines the material and methods used, including a theoretical and empirical approach that identifies the harmony existence and relationship in agriculture, environmental, and economic growth. Section 3 displays the result followed by the discussion, while the closing part compiles and concludes.

## 2. Material and Methods

### *2.1. Theoretical Approach*

#### 2.1.1. Green Growth Grants to Agriculture and Economy

Green growth strategies promote environmental externalities favorable for agricultural production and resource management, as they increase carbon sequestration and improve the natural soil matter to yield tangible benefits [21]. Moreover, the green sector becomes a crucial job creation boosting the national incomes [22]. Due to its geographic location and resource availability, there are significant economic benefits for a developing country such as Rwanda to follow a low carbon resource-efficient pathway. This pathway provides significant potential investments for income and environmental benefits [23]. Moreover, the current Rwandese moderate urbanization, similar to other developing nations, allows in advance the future planning towards greened livable cities for sustainable development [24].

#### 2.1.2. Agricultural Grants to Economy and Environment

As of 2018, agriculture served only 3% of the world's economy, dropping from 4% in 2010; despite that small portion, the sector employs almost 30% of the worldwide workers, especially in developing nations [25]. In Rwanda, agriculture contributes 28% of the GDP as of 2018. Thus, an improvement of that economic contributor is not only for social benefit but also for economic welfare [12]. The sector produces a series of environmental assistance necessary for green growth, including greenhouse gas mitigation. The Intergovernmental Panel on Climate Change (IPCC) shows that, although the total direct emissions of agriculture (including livestock and animal species) are about 10–12% of the total global emissions, vegetable, and soil carbon sequestration has the potential to neutralize around 20% of the global greenhouse gases, including $CO_2$ [26]. Thus, the sector mitigates globally at least 10% of the greenhouse gas emissions. The agriculture area estimates more than 37.7% of the global total land use; adding the forest cover, this rate becomes 68.4%. Moreover, some particular cases like Wang et al. (2015) on the carbon sequestration resulted from the black locust tree, also prove that sequestration scenario [27]. Therefore, the sector performs ecosystem conservation crucial role. The land sector coverage influences the universe by controlling the water and land resources, i.e., plant species, animal habitats, flood control, woodlands, biodiversity preservation, and landscape protection [28]. Figure 1 and the above statements enhance how green growth policies are the intercessor of economic growth, several economic activities, and sustainable development.

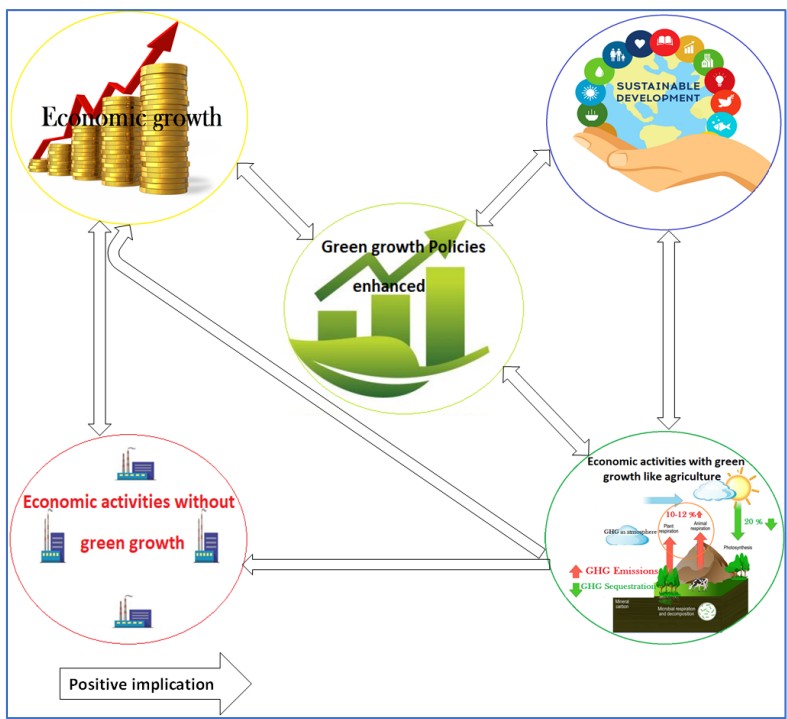

**Figure 1.** Sustainability, economic activities, economy, and green growth harmony.

### 2.1.3. Status of Green and No Green Growth Sectors in Rwanda

Green growth activities, including agriculture, are the country's principal economic components offered on an average of 76.4% of the GDP, while the no green sectors gave on average 23.6% in 2006–2018. Only the agriculture sector share was averaging 29% and hired 75.1% of the workforce in that period (Tables 1 and 2). However, the agricultural sector decreased from 33% to 28% in 2006–2018 (Figure 2) and mainly comprised of the elderly population with low education levels (Figure 3).

**Table 1.** Economic contribution proportion by sector from 2006 to 2018.

| Sector's Shares (%) | 2006 | 2008 | 2010 | 2012 | 2014 | 2016 | 2018 | Average |
|---|---|---|---|---|---|---|---|---|
| Agriculture | 33 | 29 | 28 | 29 | 28 | 30 | 28 | 29 |
| Green growth services | 44 | 48 | 47 | 47 | 48 | 47 | 48 | 47.4 |
| No green growth industry | 16 | 15 | 17 | 18 | 17 | 16 | 17 | 16.5 |
| Taxes | 7 | 8 | 8 | 6 | 7 | 7 | 7 | 7.1 |

Source: National Institute of Statistics of Rwanda (NISR) and author's calculations.

**Table 2.** Agriculture employment, education levels, and age group proportions 2002–2018.

| Years | 2002 | 2006 | 2010 | 2014 | 2018 | Average |
|---|---|---|---|---|---|---|
| Employment Proportion % | 86.6 | 79 | 72.5 | 67.6 | 69.8 | 75.1 |
| **Highest level of education** | | | | | | |
| Education levels % | 2002 | 2006 | 2010 | 2014 | 2018 | Average |
| Primary, unknown & none | 96.1 | 96.6 | 96.6 | 90.1 | 94.4 | 94.76 |
| Post primary | 2.3 | 1.8 | 1.1 | 7 | 5.3 | 3.50 |
| Secondary & vocational | 1.6 | 1.5 | 2.1 | 2.7 | 0.3 | 1.64 |
| University | 0 | 0.1 | 0.2 | 0.2 | 0.0 | 0.10 |
| **Age group** | | | | | | |
| Age group | 16–24 | 25–34 | 35–44 | 45–54 | 55–64 | 65+ | Average |
| percentages | 62.8 | 64.2 | 68.4 | 78.1 | 85.1 | 92.0 | 75.1 |

Source: NISR and author's calculations.

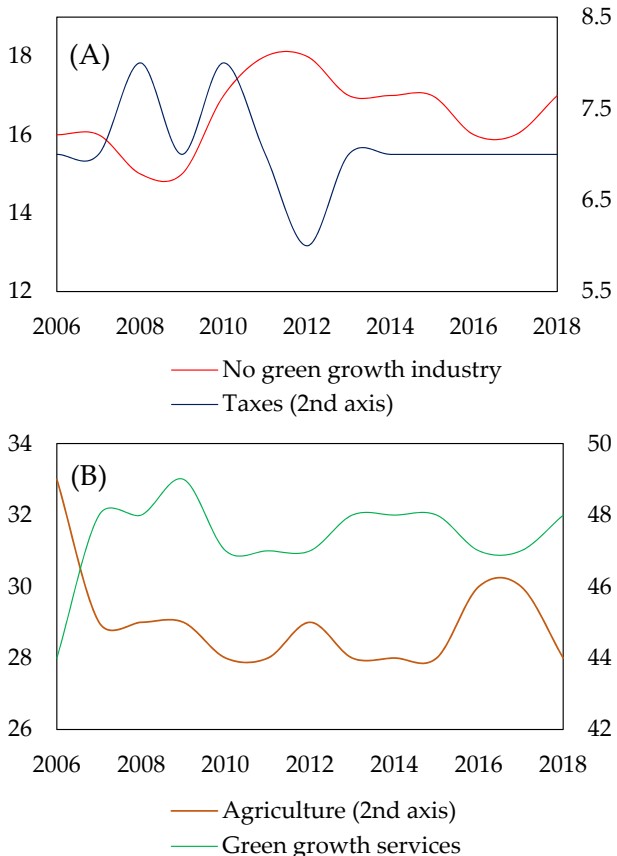

**Figure 2.** Shares to GDP of no green growth (**A**), and green growth (**B**) economic activities.

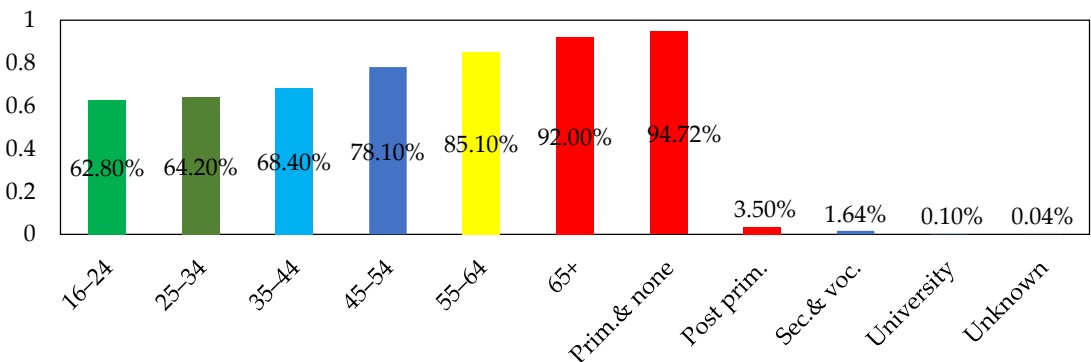

**Figure 3.** Agriculture employment age and education level proportions.

*2.2. Description of Variables*

This research applied data from the World Bank dataset 2019, the National Institute of Statistics of Rwanda (NISR), the Ministry of agriculture, and Rwanda Environmental Management Authority (REMA). It traced the 132 monthly $CO_2$, AGRc, and GDPc time series data for the January 2008– December 2018 period, as the green economy enhancement period. Variables structures are presented in Figure 4.

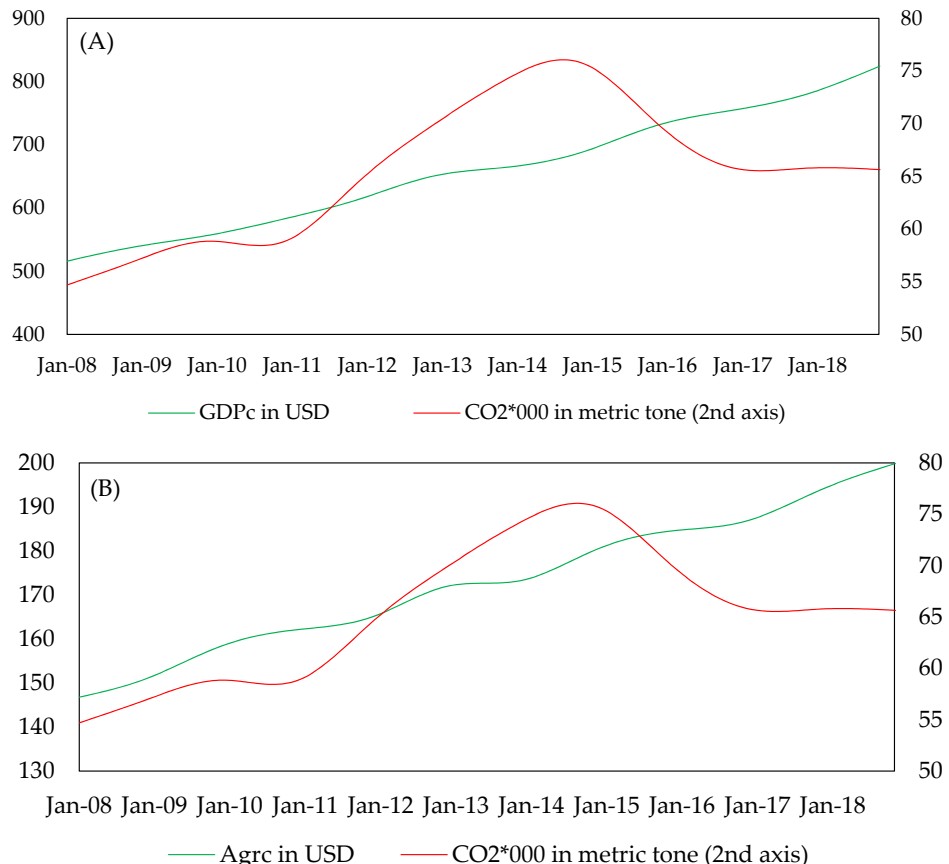

**Figure 4.** Time series plots: GDPc-$CO_2$ (**A**), and AGRc-$CO_2$ (**B**) January 2008−December 2018.

### 2.3. Model Specification

The aggregated functional form defining the relationship between $CO_2$, GDPc, and AGRc is:

$$LCO_2(000) = function(LGCF_c, \ LAGR_c) \tag{1}$$

Reflecting on the early analysis a decade ago of [29,30], and the recent with the improved algorithm, including [31–33], an appropriate model from Equation (1) is discovered with the following alternative equations:

$$LCO_2(000) = \alpha_0 + \alpha_1 LGDPc + \alpha_3 LAGRc + u_i \tag{2}$$

$$LCO_2(000) = \alpha_0 + \alpha_1 LGDPc + \alpha_2 LGDPc^2 + \alpha_3 LAGRc + u_i \tag{3}$$

$$LCO_2(000) = \alpha_0 + \alpha_1 LGDPc + \alpha_3 LAGRc + \alpha_4 LAGRc^2 + u_i \tag{4}$$

$$LCO_2(000) = \alpha_0 + \alpha_1 LGDPc + \alpha_2 LGDPc^2 + \alpha_3 LAGRc + \alpha_4 LAGRc^2 + u_i \tag{5}$$

$LCO_2$, LGDPc, and LAGRc are the natural log transform of $CO_2$, GDPc, and AGRc, respectively. $\alpha_1$, $\alpha_2$, $\alpha_3$, and $\alpha_4$ are the parameters indicating the $CO_2$ long and short-term elasticity the regressors. $\alpha_0$ is the intercept, while $u_i$ is the error term observed as independent and normally distributed with mean zero and constant variance.

Following the EKC or GBMLQ hypothesis, the turning point occurs alternatively or simultaneously at the agriculture production level of $\alpha_3/2\alpha_4$ and/or income level of $\alpha_1/2\alpha_2$. It means that the $CO_2$ emissions level rise as country income and/or agriculture, then decline as rising income and/or agriculture output pass through the turning point. Figure 5 summarizes the model diagnostics, and testifies the EKC adoption in developing countries.

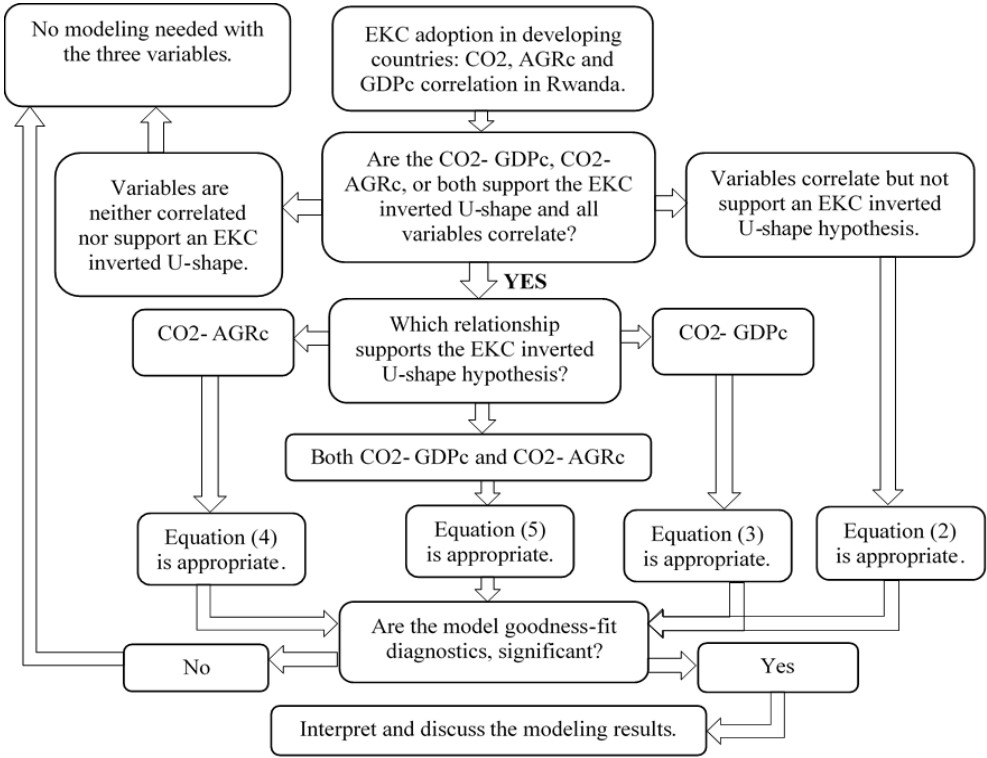

**Figure 5.** Model summary processes testifying the environmental Kuznets curve (EKC) adoption and diagnostics.

### 2.3.1. Tests for the EKC or GBMLQ Inverted U-Shape Hypothesis

Two tests were carried for an inverted U-shape pattern. First, the opposite signs of $\alpha_1$ to $\alpha_2$, and $\alpha_3$ to $\alpha_4$ or decoupling and coupling scenarios [34]. The coupling and decoupling concept were denoted in terms of $CO_2$ elasticity from GDPc, and AGRc. If one of the variables increase as the $CO_2$ goes up, it is a positive $CO_2$ elasticity. If that positive elasticity is less than unity, the variable increases lower quickly than $CO_2$, indicating a relative decoupling. For absolute coupling relationship, elasticity is positive and greater nor equal to unity, and regressors increases at the same or large rapidity as the $CO_2$ (Equation (6)).

$$\text{Relative decoupling}: \frac{\delta LCO_2}{\delta LGDPc} = g'(LGDPc) < 1$$

$$\text{Absolute decoupling}: \frac{\delta LCO_2}{\delta LGDPc} = g'(LGDPc) < 0$$

$$\text{Relative coupling}: \frac{\delta LCO_2}{\delta LGDPc} = g'(LGDPc) > 0 \tag{6}$$

$$\text{Absolute coupling}: \frac{\delta LCO_2}{\delta LGDPc} = g'(LGDPc) > 1$$

If the parameters' signs of linear and quadratic terms are the same, they reveal a monotonic absolute decoupling or coupling relationship among variables, indicating a non-validity of the hypothesis. Otherwise with the optimization rules [33,35], at $\alpha_1/2\alpha_2$ GDPc level, and/or $\alpha_3/2\alpha_4$ AGRc level; the turning points occurred when:

$$\frac{\delta LCO_2}{\delta LGDPc} = 0 \text{ iff } LGDPc = \alpha_1/2\alpha_2$$

$$\frac{\delta LCO_2}{\delta LAGRc} = 0 \text{ iff } LAGRc = \alpha_3/2\alpha_4 \tag{7}$$

Second, the R$^2$-adjusted significance produced from the added quadratic terms: To avoid overeating, counterfeit, or both in the regression results, if adding the quadratic term resulted in an adjusted R$^2$ that is less than the one in a monotonic equation, it implied a non-validity of the hypothesis. Otherwise, the hypothesis is appropriate [31]. Thus, the modeling analysis was used to define the appropriate Equation either (2), (3), (4), or (5).

### 2.3.2. ARDL Method for Cointegration

The cointegration precondition is the presence of unit root. Unit roots and variables integration order tests were applied, including Peron Phillips (PP) [36] and Augmented dickey fuller (ADF) [37]. ARDL model contains the lagged value (s) of the dependent variable and the regressors' current and lagged values as independent variables. If variables are cointegrated from the bounds test, both the long run-VECM and short-run-ARDL models are designated; otherwise, only the short run-ARDL. The long-term and cointegration between variables is the regular OLS regression with the lags of explained and explanatory variables, expressed as ARDL (p, k1, k2 . . . ki). Where p is the explained variable lags number and ki the number of the explanatory variables. A magnetic error correction term is generated in cointegration, connecting the short-run causality in the long run without latter information loss.

The cointegration, EKC, ECM and ARDL approaches have been used by several researchers on economic and green growth variables, including [38], and the recent [39,40]. ARDL is superior to the classical methods, considering its convenience of not reflecting on the variable's integration of the same order as it can be fitted when variables are at I (0), I (1), or jointly, but not at I (2) or more. It is also more effective in the small and limited sample sizes [32], as is in this research. Thus, the research results from this study could be unbiased. However, as other methods analyzed the secondary data, our methodology's features could not capture or specify the population or stakeholders' suggestions and perceptions on the green sector's investments.

### 2.3.3. Model Estimation

The generalized ARDL (P, q) is displayed as vectors in Equation (8):

$$Y_t = Y_0 + \sum_{i=1}^{p} \delta_i Y_{t-i} + \sum_{i=1}^{q} \beta_i X_{t-i} + \varepsilon_{it} \tag{8}$$

where: $Y_t$ and $X_i$ the dependent and independent variables, $\delta$ and $\beta$ the variables coefficients, $Y_0$ the intercept, p and q the dependent and independent variables lag, and $\varepsilon_{it}$ the error terms. The explained variable depends on its lagged values, and the current and lagged values of explanatory variables. Thus, the ARDL (p, $q_1$, $q_2$, $q_3$, $q_4$) cointegration bounds test is:

$$
\begin{aligned}
LCO_2(000) = \ & a_{01} + b_{11}LCO_2(000)_{t-i} + b_{12}LGDPc_{t-i} + b_{13}LGDPc^2_{t-i} + b_{14}LAGRc_{t-i} \\
& + b_{15}AGRc^2_{t-i} + \sum_{i=1}^{p} a_{i1}LCO_{2t-i} + \sum_{i=1}^{q} a_{i2}LGDPc_{t-i} \\
& + \sum_{i=1}^{q} a_{i3}LGDPc^2_{t-i} + \sum_{i=1}^{q} a_{i4}LAGRc_{t-i} + \sum_{i=1}^{q} a_{i5}LAGRc^2_{t-i} + e_{t1}
\end{aligned}
\tag{9}
$$

The no cointegration null hypothesis is revealed if $b_{i1} = b_{i2} = b_{i3} = b_{i4} = b_{i5}$, the alternative of cointegration occurs if $b_{i1} \neq b_{i2} \neq b_{i3} \neq b_{i4} \neq b_{i5}$, where I = 1 to 5. Equation (8) expresses $Y_t$ as a vector; thus, to test the cointegration, each of the five variables is used as the dependent.

If no cointegration, only the short-term ARDL (p, $q_1$, $q_2$, $q_3$, $q_4$) model is specified as:

$$
\begin{aligned}
\Delta \mathrm{LCO_2(000)} = \quad & a_{01} + \sum_{i=1}^{p} a_{i1} \Delta \mathrm{LCO_2(000)}_{t-i} + \sum_{i=1}^{q1} a_{i2} \Delta \mathrm{LGDPc}_{t-i} \\
& + \sum_{i=1}^{q2} a_{i3} \Delta \mathrm{LGDPc}^2_{t-i} + \sum_{i=1}^{q3} a_{i4} \Delta \mathrm{LAGRc}_{t-i} + \sum_{i=1}^{q4} a_{i5} \Delta \mathrm{LAGRc}^2_{t-i} + e_t
\end{aligned}
\tag{10}
$$

If cointegration exists, the ARDL long and short run respectively estimates are:

$$
\begin{aligned}
\mathrm{LCO_2(000)} = \quad & a_{01} + \sum_{i=1}^{p} a_{i1} \mathrm{LCO_2(000)}_{t-i} + \sum_{i=1}^{q1} a_{i2} \mathrm{LGDPc}_{t-i} + \sum_{i=1}^{q2} a_{i3} \mathrm{LGDPc}^2_{t-i} \\
& + \sum_{i=1}^{q3} a_{i4} \mathrm{LAGRc}_{t-i} + \sum_{i=1}^{q4} a_{i5} \mathrm{LAGRc}^2_{t-i} + e_{t1}
\end{aligned}
\tag{11}
$$

$$
\begin{aligned}
\Delta \mathrm{LCO_2} = a_{01} + \quad & \sum_{i=1}^{p} a_{i1} \Delta \mathrm{LCO_2(000)}_{t-i} + \sum_{i=1}^{q1} a_{i2} \Delta \mathrm{LGDPc}_{t-i} + \sum_{i=1}^{q2} a_{i3} \Delta \mathrm{LGDPc}^2_{t-i} \\
& + \sum_{i=1}^{q3} a_{i4} \Delta \mathrm{LAGRc}_{t-i} + \sum_{i=1}^{q4} a_{i5} \Delta \mathrm{LAGRc}^2_{t-i} + \lambda \mathrm{ECT}_{t-1} + e_{t2}
\end{aligned}
\tag{12}
$$

where $\mathrm{ECT}_{t-1}$: the error correction term from the cointegrated Equation (9), showing the variables quickness convergence to the equilibrium, $\lambda$: the adjustment speed for every shock attempting the long-term equilibrium deviation ($\lambda$ comes with a negative sign to prove the model convergence in the long term; otherwise, the model is explosive). $e_t$: serial uncorrelated error term. From Equations (11) and (12) we have:

$$
\begin{aligned}
\mathrm{ECT_t} = \mathrm{LCO_2} - \quad & a_{01} - \sum_{i=1}^{p} a_{i1} \mathrm{LCO_2(000)}_{t-i} - \sum_{i=1}^{q1} a_{i2} \mathrm{LGDPc}_{t-i} - \sum_{i=1}^{q2} a_{i3} \mathrm{LGDPc}^2_{t-i} \\
& - \sum_{i=1}^{q3} a_{i4} \mathrm{LAGRc}_{t-i} - \sum_{i=1}^{q4} a_{i5} \mathrm{LAGRc}^2_{t-i}
\end{aligned}
\tag{13}
$$

The VECM is established if all the alternative equations are cointegrated. Cointegration exists if the bound test F-statistic is greater than the upper critical bound value for the test. If the value is between the minimum and maximum values, there is no conclusion. There is no cointegration in the model if the F-statistic is less than the lower critical bound [41].

### 2.3.4. Causality

Causality analysis is based on the ARDL-VECM; lagged forms define the variables short term, long term, and joint causality. The short-run explanatory variables t and $\chi^2$ statistics perform the short-run causal, $\lambda$ captures the long-run causality, while the regressors F-statistic and t-statistic of the ECT are for joint causal [32]. Cointegration indicates the existence of the causality at least one direction. Once confirmed, the next step is to determine the causality direction, using the ECT identified by the long run VECM.

$$
\begin{bmatrix} \Delta LCO_{2t} \\ \Delta LGDPc_t \\ \Delta LGDPc^2{}_t \\ \Delta LAGRc_t \\ \Delta LAGRc^2{}_t \end{bmatrix} = \begin{bmatrix} \alpha_1 \\ \alpha_2 \\ \alpha_3 \\ \alpha_4 \\ \alpha_5 \end{bmatrix} + \begin{bmatrix} \delta_{11,1} & \delta_{12,1} & \delta_{13,1} & \delta_{14,1} & \delta_{15,1} \\ \delta_{21,1} & \delta_{22,1} & \delta_{23,1} & \delta_{24,1} & \delta_{25,1} \\ \delta_{31,1} & \delta_{32,1} & \delta_{33,1} & \delta_{34,1} & \delta_{35,1} \\ \delta_{41,1} & \delta_{42,1} & \delta_{43,1} & \delta_{44,1} & \delta_{45,1} \\ \delta_{51,1} & \delta_{52,1} & \delta_{53,1} & \delta_{54,1} & \delta_{55,1} \end{bmatrix} \begin{bmatrix} \Delta LCO_{2t-1} \\ \Delta LGDPc_{t-1} \\ \Delta LGDPc^2{}_{t-1} \\ \Delta LAGRc_{t-1} \\ \Delta LAGRc^2{}_{t-1} \end{bmatrix} + \ldots
$$
$$
+ \begin{bmatrix} \delta_{11,k} & \delta_{12,k} & \delta_{13,k} & \delta_{14,k} & \delta_{15,k} \\ \delta_{21,k} & \delta_{22,k} & \delta_{23,k} & \delta_{24,k} & \delta_{25,k} \\ \delta_{31,k} & \delta_{32,k} & \delta_{33,k} & \delta_{34,k} & \delta_{35,k} \\ \delta_{41,k} & \delta_{42,k} & \delta_{43,k} & \delta_{44,k} & \delta_{45,k} \\ \delta_{51,k} & \delta_{52,k} & \delta_{53,k} & \delta_{54,k} & \delta_{55,k} \end{bmatrix} + \begin{bmatrix} \lambda_1 \\ \lambda_2 \\ \lambda_3 \\ \lambda_4 \\ \lambda_5 \end{bmatrix} ECT_{t-1} + \begin{bmatrix} e_{t1} \\ e_{t2} \\ e_{t3} \\ e_{t4} \\ e_{t5} \end{bmatrix}
$$

(14)

Equations (9) and (12) generate an expanded causal test concerning ECT by the combined VECM, where $e_{t1}$ to $e_{t5}$ are residual, and $\delta$ the short-run elasticity parameters (Equation (14)).

### *2.4. Model Diagnostics*

Using the Jarque–Bera (JB) test, the goodness–fit was verified through the heteroscedasticity, serial correlation, Ramsey function, and residuals normality tests. The model stability was tested based on residuals recursive, verified by the Cumulative Sum of Recursive Residuals (CUSUM), represented graphically by two straight lines bounded by the significance level. If the plots slump within the critical significance bounds, the generated regression coefficients are stable, and the null hypothesis of stable coefficients is accepted [4].

## 3. Data Analysis and Discussion

### *3.1. Data Analysis*

#### 3.1.1. Summary Statistics

This research used annual data on agriculture production, $CO_2$ emissions, and the real GDP from 2008 to 2018. $CO_2$ emissions are calculated in metric tons, while the GDPc and agriculture production are in US dollars at 2010 prices (Table 3).

**Table 3.** Statistics of variables in summary, 2008–2018.

| Variables Statistics | GDPc | AGRc | CO$_2$ (000) |
|---|---|---|---|
| Mean | 671.2521 | 175.2087 | 66.0352 |
| Standard deviation | 93.9015 | 15.3508 | 6.1827 |
| Coefficient of variation CV (%) | 13.9890 | 8.7614 | 9.3627 |

Source: author's calculations.

Figure 4 shows the change in Rwanda's GDPc-$CO_2$ and AGRc-$CO_2$ trends. GDPc and AGRc variables increased with $CO_2$ up to the EKC turning point, then $CO_2$ trends negatively. However, the agriculture sector's coefficient of variation (CV) is at a low rate (Table 3).

Table 4 compares Rwanda with the neighboring developing nations in the same East African Community (EAC) regional integration, where green growth is not yet adopted (Figure 6).

In (2013–2018) period, the Rwandese AAGR and CAGR of GDPc rounded 4.4%, almost 1.42 times higher than the Kenyan, the best economic performer in EAC, the AAGR, and CAGR rates of AGRc were 2.9%, approximately 1.45 times higher than the regional best performer country. Moreover, Rwanda's $CO_2$ growth rate was reduced by around 2.3%, while, in all other EAC countries, it raised by around 18%, 2%, 6%, and 2% for Burundi, Kenya, Tanzania, and Uganda, respectively. Despite the Rwandese GDPc growth with emission reduction (Figure 6), the AGRc growth rates remained almost

1.5 times less than the GDPc. Therefore, for the last 10 years, the increase of Rwandese green growth activities made significant gains in CO2 emissions reduction with a high increment of the GDPc than AGRc (Table 4).

**Table 4.** Variable average and compound annual growth rates in East African Community (EAC) 2008–2018.

| Country | Years | Variables | Average Annual Growth Rates (AAGR in %) | Compound Annual Growth Rates (CAGR %) |
|---------|-------|-----------|------------------------------------------|----------------------------------------|
| Rwanda | (2008–2013) | GDPc | 4.30 | 4.29 |
| | | AGRc | 2.79 | 2.78 |
| | | $CO_2$ (000) | 5.50 | 5.45 |
| | (2013–2018) | GDPc | 4.42 | 4.41 |
| | | AGRc | 2.90 | 2.89 |
| | | $CO_2$ (000) | −2.31 | −2.38 |
| Burundi | (2008–2013) | GDPc | 1.17 | 1.17 |
| | | AGRc | −1.29 | −1.35 |
| | | $CO_2$ (000) | 5.48 | 5.29 |
| | (2013–2018) | GDPc | −2.76 | −2.79 |
| | | AGRc | −4.18 | −4.20 |
| | | $CO_2$ (000) | 18.80 | 17.59 |
| Kenya | (2008–2013) | GDPc | 2.85 | 2.84 |
| | | AGRc | 0.97 | 0.90 |
| | | $CO_2$ (000) | 3.04 | 2.62 |
| | (2013–2018) | GDPc | 3.09 | 3.09 |
| | | AGRc | 2.02 | 2.01 |
| | | $CO_2$ (000) | 2.09 | 1.92 |
| Tanzania | (2008–2013) | GDPc | 3.04 | 3.04 |
| | | AGRc | 0.21 | 0.21 |
| | | $CO_2$ (000) | 9.79 | 9.45 |
| | (2013–2018) | GDPc | 3.22 | 3.21 |
| | | AGRc | 2.64 | 2.64 |
| | | $CO_2$ (000) | 5.69 | 5.59 |
| Uganda | (2008–2013) | GDPc | 2.53 | 2.51 |
| | | AGRc | −0.86 | −0.87 |
| | | $CO_2$ (000) | 5.73 | 5.41 |
| | (2013–2018) | GDPc | 1.31 | 1.31 |
| | | AGRc | −0.97 | −0.97 |
| | | $CO_2$ (000) | 1.68 | 1.60 |

Source: author's calculations.

**Figure 6.** *Cont.*

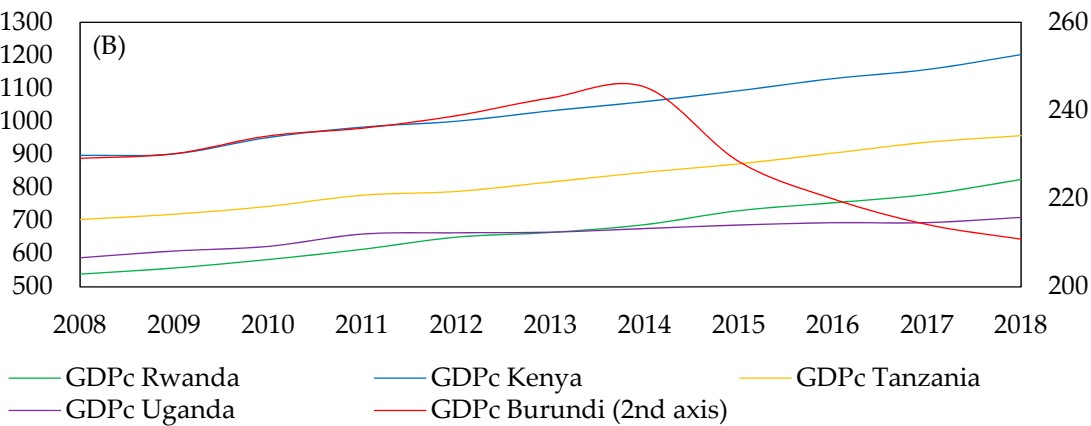

**Figure 6.** Time series plots 2008–2018; EAC countries $CO_2$ (000) (**A**), and GDPc (**B**).

### 3.1.2. Lag Selection

The variables interdependency is rarely immediate; X reacts to Y with a lapse of time called lag. Extra lags cause degrees of freedom loss, leading to statistically insignificant and can cause multicollinearity. Few lags lead to misspecification errors. Randomly: for annual data, the lag is 1 to 2, quarterly 1 to 8, and monthly 1 to 24 [42]. In this research, the preferred lag length is one. The chosen lag in this paper is one (Table 5) (* shows the chosen lag by HQ: Hannan-Quinn information criterion, LR: sequentially modified LR test statistic, FPE: Final prediction error, AIC: Akaike information criterion, and SC: Schwarz information criterion).

**Table 5.** Variables lag length.

| Lag | LogL | LR | FPE | AIC | SC | HQ |
|---|---|---|---|---|---|---|
| 0 | 232.1130 | NA | 0.001799 | −3.482642 | −3.394849 | −3.446968 |
| 1 | 554.3840 | 619.9412 * | $1.33 \times 10^{-5}$ * | −8.387543 * | −8.277802 * | −8.342950 * |
| 2 | 156.0872 | 65.3826 | $2.64 \times 10^{-17}$ | −27.2174 | −26.6123 | −27.8813 |

Source: author's calculations.

### 3.1.3. Cointegration Test

The bounds test ARDL is appropriate if the descriptive variables are at I (0), I (1) or mutually, and notably none of the variables is at I (2) or above.

ADF and PP tests are consistent with no stationary variables (unit root) at the level, and stationary at I (1). Hence, the ARDL bounds test for cointegration could proceed (Table 6).

**Table 6.** The variables unit-roots tests.

| Tests | ADF | | PP | |
|---|---|---|---|---|
| | Level | 1st Diff. | Level | 1st Diff. |
| $LCO_2$ (000) | −2.19 | −1.54 * | 0.61 | −1.62 * |
| LGDPc | −0.0026 | −4.51 ** | 0.40 | −4.77 *** |
| $LGDPc^2$ | 0.18 | −4.87 *** | 0.91 | −5.05 *** |
| LAGRc | 0.06 | −6.48 *** | −1.24 | −14.13 *** |
| $LAGRc^2$ | 0.21 | −6.51 *** | −0.93 | −13.00 *** |

***, **, * indicates the $H_0$ rejection of a unit root in the series at 1%, 5%, and 10%, respectively. Source: author's calculations.

### 3.1.4. Relevant Equation for the Model

The appropriated equation among (2)–(5) is established through the verification of EKC inverted U-shape hypothesis and cointegration bounds test (Figure 6).

Adding a quadratic term $LGDPc^2$ on the linear form results on an adjusted $R^2$ greater than 10% and higher than other alternatives options in differences; thus, Equation type (3) is (Table 7). The turning point of inverted U-shape is at around an income level of 6.59 in natural logarithm ($=50.65348/(2 \times 3.842931)$) in December 2015. Therefore, only the $CO_2$-GDPc relationship describes an EKC inverted U-shape (Figure 7), while the $CO_2$–AGRc relationship presents a monotonic, where the AGRc increases at the expenses of $CO_2$.

**Table 7.** $CO_2$ relevant equation type selection.

| Equation | Independent Variables | | | | Intercept | Test Statistic | |
| | LGDPc (−1) | LGDPc$^2$ (−1) | LAGRc (−1) | LAGRc$^2$ (−1) | | R$^2$-Adjusted | R$^2$-Adjusted Comparison |
| --- | --- | --- | --- | --- | --- | --- | --- |
| Equation (2) | 0.145659 | | 0.516941 | | 0.575084 | 0.425584 | |
| Equation (3) | 50.65348 | −3.842931 | −0.667313 | | −159.1832 | 0.885569 | (3)-(2) = 39% |
| Equation (4) | 1.131848 | | 79.42100 | −7.821303 | −204.6764 | 0.737725 | (4)-(2) = 31.21% |
| Equation (5) | 149.9745 | −11.57183 | −183.8918 | 17.92589 | −10.06670 | 0.882306 | (5)-(3) = 6.67% |

Source: author's calculations.

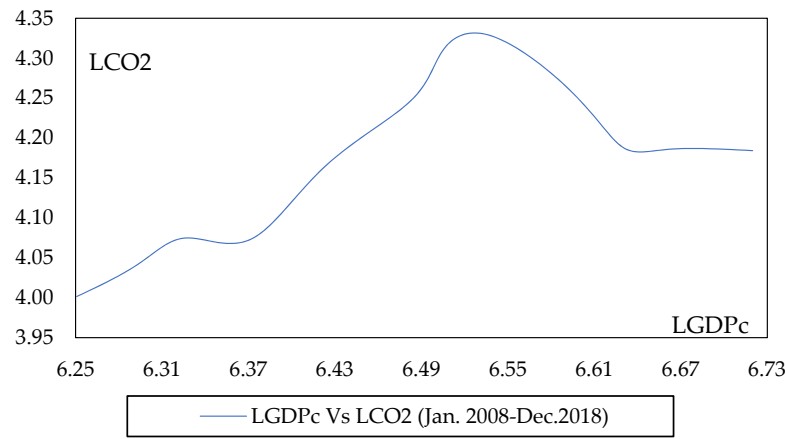

**Figure 7.** LGDPc-LCO$_2$ (000) emissions plot January 2008–December 2018.

Results show that the F-statistic equal to 11.3278 is higher than the upper bound values at 1%, 5%, and 10% levels, respectively, and alternatives functions are also cointegrated (Table 8). Thus, the no cointegration null hypothesis is rejected at all those levels. The model's R2 is 99.8% (Table 9), indicating that 99.8% of the dependent variable is explained by the regressors. The fitted and actual graphs are closely related (Figure 8).

**Table 8.** Bounds test for cointegration with relevant equation.

| Function | F-Statistic Value | Lags | Significance Levels | | | Cointegrated |
| | | | Levels | Lower Bound I (0) | Upper Bound I (1) | |
| --- | --- | --- | --- | --- | --- | --- |
| $LCO_2(000) = F$ (LGDPc, LGDPc2, LAGRc) | 11.3278 | 1, 1, 1, 1 | 10% 5% 1% | 2.72 3.23 4.29 | 3.77 4.35 5.61 | Yes |
| $LGDPc = F(LCO_2(000), LGDPc2, LAGRc)$ | 916.1651 | 1, 1, 1, 1 | 10% 5% 1% | 2.72 3.23 4.29 | 3.77 4.35 5.61 | Yes |
| $LGDPc^2 = F(LCO2(000), LGDPc, LAGRc)$ | 921.6545 | 1, 1, 1, 1 | 10% 5% 1% | 2.72 3.23 4.29 | 3.77 4.35 5.61 | Yes |
| $LAGRc = F(LCO2(000), LGDPc, LGDPc2)$ | 6.0842 | 1, 1, 1, 1 | 10% 5% 1% | 2.72 3.23 4.29 | 3.77 4.35 5.61 | Yes |

Source: author's calculations.

**Table 9.** Results of the model diagnostic tests.

| Long-Run Model Diagnostic Tests | | | | | Short-Run Model Diagnostic Tests | | | | |
|---|---|---|---|---|---|---|---|---|---|
| Residual diagnostics | Serial Correlation | Breusch-Godfrey LM test | $R^2$ observed | 3.0980 ** (0.0784) | Serial Correlation | Breusch-Godfrey LM test | $R^2$ observed | 7.0933 (0.0077) |
| | | Q-statistic probability | Q-stat. | 0.3570 ** (0.5500) | | Q-statistic probability | Q-stat. | 4.7167 * (0.0300) |
| | Heteroskedasticity | Breusch-Pagan test | $R^2$ observed | 5.2116 ** (0.3654) | Heteroskedasticity | Breusch-Pagan test | $R^2$ observed | 6.0445 *** (0.3018) |
| Stability diagnostics | Miss specification test | Ramsey's test | t-statistic | 1.9555 *** (0.1222) | Miss specification test | Ramsey's test | t-statistic | 0.5397 *** (0.6435) |
| | | | F-statistic | 3.8243 *** (0.1222) | | | F-statistic | 0.2912 *** (0.6435) |
| | Recursive estimation | CUSUM | 5% level | Results within 5% level | Recursive estimation | CUSUM | 5% level | Results within 5% level |

*, **, and *** are the no rejection of the null hypothesis of no serial correlation, no heteroskedasticity, or no model miss specification presence at 10%, 5%, and 1% levels of significance, respectively, where the () values are the *p*-values. Source: author's calculations.

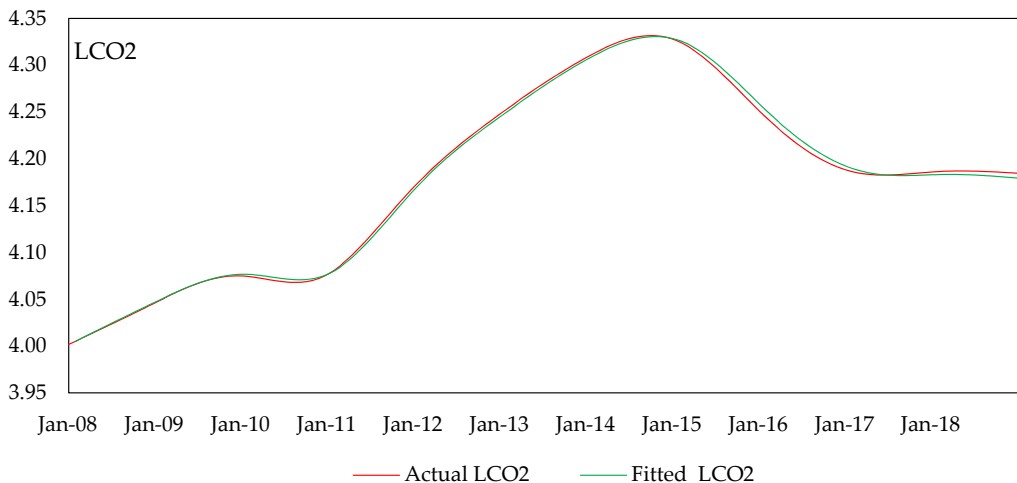

**Figure 8.** Relevant model: actual and fitted LCO$_2$ plots January 2008–December 2018.

With the EKC hypothesis, LGDPc and LGDPc$^2$ are with parameters $\alpha 1 > 0$ and $\alpha 2 < 0$ in both long term and short-term connections. Thus, the ARDL (1, 1, 1, 1) long-run cointegrated equation is:

$$
\begin{aligned}
\text{LCO}_2(000) = -5.9431 + 0.9778\text{LCO}_2(000)(-1) + 2.0075\,\text{LGDP}_c(-1) \\
-0.1486\text{LGDP}_c^2(-1) - 0.1414\text{LAGR}_c(-1) + u_i
\end{aligned}
\tag{15}
$$

Thus;

$$
\begin{aligned}
\text{LCO}_2 = -0.0059431 + 0.9972\text{LCO}_2(-1) + 0.0020075\text{LGDP}_c(-1) \\
-0.0001486\text{LGDP}_c^2(-1) - 0.0001414\text{LAGR}_c(-1) + u_i
\end{aligned}
\tag{16}
$$

While the ARDL (1, 1, 1, 1) short-run equation is:

$$
\begin{aligned}
\Delta\text{LCO}_2(000)_t = 0.000491 + 0.9972\Delta\text{LCO}_2(000)_{(-1)} + 0.8673\Delta\text{LGDPc}_{(-1)} \\
-0.671\Delta\text{LGDPc}_{(-1)}^2 - 0.2178\Delta\text{LAGRc}_{(-1)} + u_i
\end{aligned}
\tag{17}
$$

Thus;

$$
\begin{aligned}
\Delta\text{LCO}_{2t} = 0.000000491 + 0.9972\Delta\text{LCO}_2(000)_{(-1)} + 0.0008673\Delta\text{LGDPc}_{(-1)} \\
-0.000671\Delta\text{LGDPc}_{(-1)}^2 - 0.0002178\Delta\text{LAGRc}_{(-1)} + u_i
\end{aligned}
\tag{18}
$$

### 3.1.5. Residual Diagnostics

The process includes the serial correlation and residual heteroskedasticity tests. The serial correlation explains the relationship between the same variable perceptions across precise periods. If it is zero, there is no serial correlation, and the observations are independent of each other. Otherwise, there is a serial correlation where future observations are affected by past values, and the model is not entirely accurate. It occurs if the errors affiliated with a given period move over toward the coming periods [4]. As in Table 9, the no serial correlation hypothesis is accepted, confirming the independence of each observation.

Heteroskedasticity occurs when the variable's standard errors observed over a specific time are not constant, which is a violation of the regression modeling theories. In this paper, the no heteroskedasticity null hypothesis in both long term and short-term models is accepted. Thus, the variance residual is the same in the model (Table 9).

### 3.1.6. Stability Diagnostic

Stability diagnostic includes the test for model misspecification and recursive estimation. In this research, it is done through Ramsey's test to verify whether the fitted combination values explain the

dependent variable correctly. The model is mis-designated if another function form might be better to approximate the model. The null hypothesis of no misspecification is not rejected for both long and short-run models (Table 9), reinforcing our choice of Equation (3) type. On the recursive estimation, coefficients are checked for recursive estimates through the CUSUM test (Table 9); the intuition is that the series changes in structural data may be subjected to one or more structural splits [4]. The model stabilities results are within the two straight lines bounded by the 5% level, indicating the model recursive stability (Figure 9). Thus, the given regression ensures the best model fit.

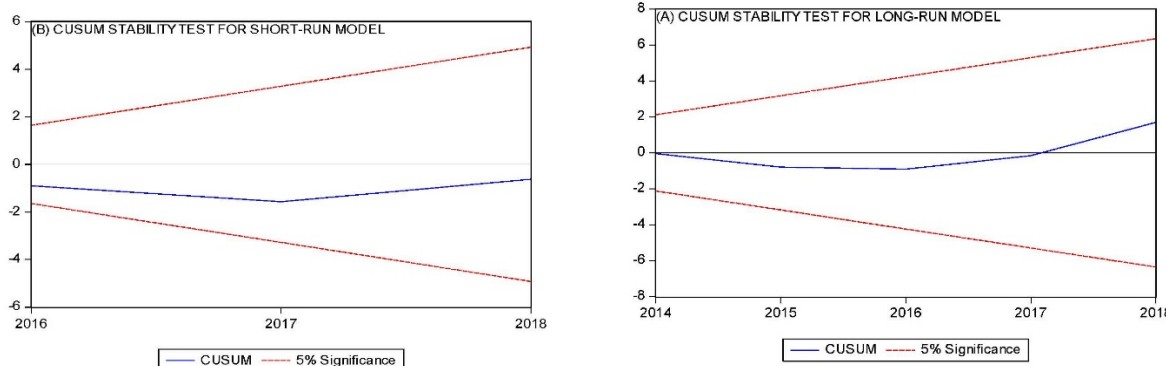

**Figure 9.** ARDL long-run (**A**) and short-run (**B**) models, CUSUM stability test results.

### 3.1.7. Causality

Causality means that the occurrence of X makes an occurrence of Y or vice versa. Thus, the cointegration indicates a causality presence in one direction, at least. It does not make a precision on its direction [33]. The ECM obtained from the long term cointegration is used for causality direction (Figure 10, Table 10).

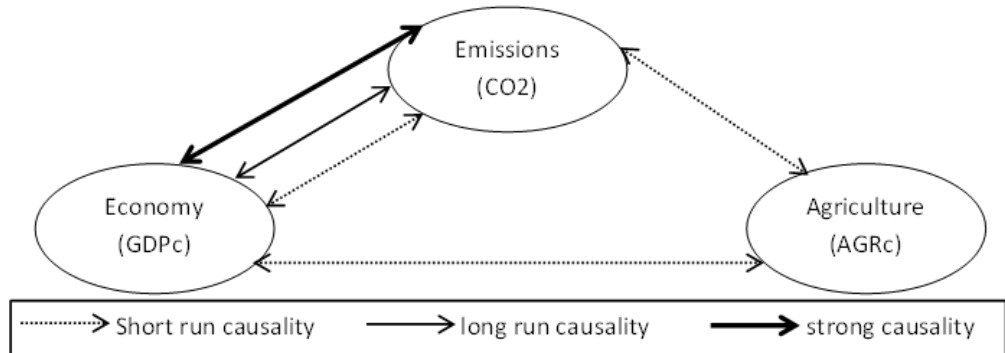

**Figure 10.** Causality relations for variables.

**Table 10.** Long-term, short-term, and VECM estimates.

| Variables | Coefficients | Std. Error | *t*-Statistic | *p* | R-Squared |
|:---:|:---:|:---:|:---:|:---:|:---:|
| | | **Long-run estimates** | | | |
| LCO$^2$(000) (−1) | 0.977798 | 0.007480 | 130.7264 *** | 0.0000 | |
| LGDPc (−1) | 2.007458 | 0.456357 | 4.398879 *** | 0.0000 | |
| LGDPc$^2$ (−1) | −0.148602 | 0.034636 | −4.290376 *** | 0.0000 | 0.9987 |
| LAGRc (−1) | −0.141488 | 0.036465 | −3.880119 *** | 0.0002 | |
| Intercept | −5.943106 | 1.438350 | −4.131891 *** | 0.0001 | |

**Table 10.** *Cont.*

| Variables | Coefficients | Std. Error | *t*-Statistic | *p* | R-Squared |
|---|---|---|---|---|---|
| **Short-run estimates** | | | | | |
| $\Delta LCO_2(000)$ (−1) | 0.997164 | 0.008312 | 119.9664 *** | 0.0000 | |
| $\Delta LGDPc$ (−1) | 0.867327 | 0.480245 | 1.806010 * | 0.0733 | |
| $\Delta LGDPc^2$ (−1) | −0.067058 | 0.036874 | −1.818578 * | 0.0714 | 0.9939 |
| $\Delta LAGRc$ (−1) | −0.217765 | 0.028477 | −7.647130 *** | 0.0000 | |
| Intercept | 0.000491 | 0.000114 | 4.294216 *** | 0.0000 | |
| **VECM estimates** | | | | | |
| $\Delta LCO_2(000)$ (−1) | 0.983190 | 0.027120 | 36.25342 *** | 0.0000 | |
| $\Delta LGDPc$ (−1) | 1.319094 | 0.963334 | 1.369301 ** | 0.0334 | |
| $\Delta LGDPc^2$ (−1) | −0.101936 | 0.074272 | −1.372482 ** | 0.0324 | 0.9939 |
| $\Delta LAGRc$ (−1) | −0.223091 | 0.030204 | −7.386247 *** | 0.0000 | |
| $ECT_{t-1}$ | −0.016356 | 0.030205 | 0.541490 ** | 0.0291 | |
| Intercept | 0.000523 | 0.000129 | 4.041930 *** | 0.0001 | |

*, **, *** are the significance of values at 1% levels. Source: author's calculations.

### 3.2. Discussion

Green growth policy allowed Rwanda to mitigate $CO_2$ emissions considerably while striving with economic emergence. Lighting from the EKC fundamental theory, the $CO_2$-GDPc short and long-run relations described are on an inverted-U shape curve, where the coefficient associated with the linear term of $CO_2$ is positive, while its associated on the quadratic term $CO_2$ is negative. Thus, the Rwandese $CO_2$-GDPc relationship supports the EKC theory [43], mostly applicable for developed countries, where the $CO_2$ would increase within the income growth first phase and decrease after achieving a specific point (Figure 7).

Agricultural sector, as one of the green economy alternative sectors, is not only an economic growth contributor but also a green growth promoter for the ecosystem conservation scenarios, including the sequestration of around 10% of the greenhouse gases. Outcomes reveal that the EKC adoption in developing countries with green growth policy like in Rwanda where the agriculture sector is among the principal economic components is feasible, despite its several weaknesses, including shares decrement to GDP, and workforce weaknesses (Table 2).

From 2013 to 2018, the Rwandese, growth rates compared to the neighboring countries in GDPc and AGRc occurred almost 1.42 and 1.45 times higher, respectively, than the region economic best performer. Emissions growth rate reduced by around 2.3%, while in all other regional integrated EAC countries, that growth rates increased. Thus, with a green growth policy, Rwanda shows income growth and environmentally protected significantly (Figure 6). The 2008–2018 Rwandese GDPc coefficient of variation (CV) was 14%, greater than the ones for all other EAC countries with no reinforced green policies (Table 3). It means that the green policy reinforcement in developing countries does not imply a decrease in income. Instead, when it is well managed with alternative green economy activities, it generates an economic boosting.

With lag one, data structure, and cointegration conditions (Table 6), the ARDL is the appropriate model for this research with variables short-run and short-run long-run correlations. Furthermore, the elaborated tests reveal that the estimated values support an EKC hypothesis at the $CO_2$-GDPc relationship, where the respective long run and short run appropriate equations are (16) (18). Explained as holding other terms constant: In both long and short-run relationships, the GDPc provides a positive and negative elasticity, to the $CO_2$, while the AGRc-$CO_2$ is a monotonic negative relationship (Table 10). In the long run, before the EKC turning point, the 1% rose in GDPc results on 0.0020075% increment of the per capita $CO_2$. After the EKC turning point, the per capita $CO_2$ declined at 0.0001486% as the GDPc increased at 1%. For monotonic AGRc-$CO_2$ relationship, a 1% increase in AGRc resulted

in a decline of the per capita $CO_2$ by 0.0001414%. In the short-run, the interpretation is the same using Equation (18). The $ECT_{t-1}$ coefficient is −0.016356 interpreted as the previous period (month) deviation from the long-run equilibrium is corrected in the current period at an adjustment speed of 1.64%. The model $R^2$ is 99.8% (Table 9), indicating that 99.8% of the dependent variable is explained by the regressors. The fitted and actual graphs are closely related (Figure 8).

The GDPc-$CO_2$, GDPc-AGRc, and $CO_2$-AGRc short-run bidirectional causals occurred. In the long run, only the GDPc-$CO_2$ bidirectional and strong causals exist (Table 11). Thus, if no reforms are elaborated, the current agriculture green economic situation could be inefficient in the long run not only on economic growth but also on green growth, calling for more improvements as one of the key economic and green growth sectors.

**Table 11.** Causality tests.

| Causal Source (Indep. var.) | Dep. Variables | | | | | | | | |
|---|---|---|---|---|---|---|---|---|---|
| | Short-Run Causal | | | Long Run Causal | | | Joint (Short & Long Run) | | |
| | $LCO_2$ | LGDPc LGDPc$^2$ | LAGRc | $LCO_2$ ECT | LGDPc ECT | LAGRc ECT | $LCO_2$, ECT | LGDPc, ECT | LAGRc, ECT |
| | χ2-Statistic | | | *t*-Statistic | *t*-Statistic | *t*-Statistic | F-Statistic | | |
| $LCO_2$ | | 52.14 *** | 11.98 ** | | 22.40 *** | 1.62 | | 26.07 ** | 5.99 |
| LGDPc | 74.29 *** | | 13.5 *** | 22.52 *** | | −1.63 | 37.2 ** | | 6.73 |
| LAGRc | 17.92 ** | 13.06 ** | | 1.26 | −1.28 | | 8.96 | 6.53 | |

Notes: **, and *** indicate that the null hypothesis is rejected at 5% and 1% levels, respectively. Source: author's calculations.

Similar to the literature, our results also reveal a greenhouse gas emission mitigation and economic growth improvement resulted from the green policies adopted [16,17], notably for the agricultural promotion [26,27], it occurred without even overtaken any maturity level. For the $CO_2$-GDPc relationship, after the EKC turning point, the GDPc contribute to the decline of the $CO_2$ emission, strengthening the positive outcomes of the Rwandese green policy. Nevertheless, after that point, the GDPc had a higher $CO_2$ reduction effect compared to AGRc, implying that, with the national development, the government improved facilities for the agriculture sector as other green growth activities [11]; however, contrary to other green activities, the sector's effective requirements, including investments and workforce, were still inadequate (Table 2, Figure 3). Results are also supported by Niyigaba and Peng (2020) 's outputs, where the agriculture shares to the GDP and future growth rate are decreasing [12].

## 4. Conclusions and Policy Implication

As this research showed, the EKC mostly adopted by the developed countries could also be adopted by developing nations with a green economy policy without adverse economic growth effects, and Rwanda is a leading example. The Rwandese agriculture sector is a significant alternative economic growth sector in coherence with the green growth; however, with the current situation, the sector's future performance on the economy and green growth could be inefficient.

Focusing on the research results, despite being a learning example as a developing country, the agro-economy and climate change mitigation should be reinforced, where government and partners in the sector should support the agriculture towards a sustainable status. With transformation strategies, including investment promotion, investment attractions targeting the youths through financial facilitation, appropriated farmers' training to overcome the skills gaps, and awareness of the sector's opportunities. The paper was able to identify the EKC adoption in developing countries using Rwanda's scenario. Rwandese instructions concerned with green and economic growth, especially the agriculture sector, could use the paper's results to adjust some policies and attract green investors, as the results approved its economic competitiveness. Moreover, several researchers could also base the paper's combined methodology and empirical results to verify other countries' situations or case

studies to develop more robust methodologies and academic contributions. Thus, the paper could be a building block to fill the gaps discovered by [20], using a systematic review of the green economy empirical studies that could convince and engage investors in green sectors.

The study has also identified the agriculture sector's green growth competitiveness, the issues facing the sector, and then proposed some measures to overcome those issues. Despite that, findings could neither capture the reasons for no adoption of environmental conservation policies in other developing countries, especially in the same region with Rwanda, nor the reasons for agriculture workforce abdication in Rwanda. However, such fine-grained research is beyond this paper's scope and provides research lines for further studies.

**Author Contributions:** J.N.: conceptualization, data curation, formal analysis, funding acquisition, investigation, methodology, project administration, resources, software, validation, visualization, writing—original draft, writing—review and editing. J.Y.S.: conceptualization, funding acquisition, investigation, project administration, resources, supervision, validation, visualization, writing—review and editing. D.P.: conceptualization, funding acquisition, investigation, project administration, resources, supervision, validation, visualization, writing—review and editing. C.U.: data curation, formal analysis, investigation, validation, visualization, writing—review and editing. All authors have read and agreed to the published version of the manuscript.

**Funding:** This research was funded by the Chinese Government Scholarship-Chinese University Program, established by the China Scholarship Council (CSC) of the Ministry of Education, to support Chinese universities in enrolling outstanding international students for postgraduate studies in China under the CSC No 2017GXZ024564.

**Conflicts of Interest:** The authors declare no conflict of interest.

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
