# Peer review of "Agriculture and Green Economy for Environmental Kuznets Curve Adoption in Developing Countries: Insights from Rwanda"

_sustainability, doi:10.3390/su122410381_

Round 1

Reviewer 1 Report

I was invited to review the former version of this manuscript which has been replaced by a new submission. I was quite happy of the authors changes to address my comments. I noticed that in this new version the vast majority of changes made to address my comments simply disappeared!

Therefore I decided to report you my previous comments which need to be tackled for improving your paper.

Below you can find the previous report:

The topic presented in this work is really interesting. However several challenges are strictly required to make acceptable the manuscript.

Going through some major comments:

  1. Abstract has inappropriate structure. I suggest to answer the following aspects: - general context - novelty of the work - methodology used - main results. It should be more concise. Please look at guide for authors
  2. Introduction presents interesting information, but could be improved. Section 1 should be broader, and take the topic form general perspective.  Being agriculture the most relevant sector of the bioeconomy, I would suggest the authors to add at least a paragraph where introducing the need also for bioeconomy development.

Some references to start with:

Enabling a sustainable and prosperous future through science and innovation in the bioeconomy at Agriculture and Agri-Food Canada

Transitioning towards the bio-economy: Assessing the social dimension through a stakeholder lens

Transition to a bioeconomy: Perspectives from social sciences

Towards a sustainable forest-based bioeconomy in Italy: Findings from a SWOT analysis

Enabling a transformation to a bioeconomy in New Zealand

  1. Methodology is unclear. Initially a short resume can be proposed to explain several steps. It should be better linked to the existing literature. what is its potential? its limit?
  2. Results are not always linked to the methodology. Please define the relationship and relate your finding with the relevant literature.
  3. Conclusion are succinct. Please provide practical implications, limitations and future research needs.

Reviewer 2 Report

The authors of the paper Agriculture and green economy for Environmental Kuznets curve adoption in developing countries: Insights from Rwanda presents a relevant theme, namely the adoption of green growth policies in Rwanda in sectors of activity (including agriculture) in the current context of development and climate change and on the basis of short-term and long-term cointegration of variables.

Concepts, bibliographic sources and citations are adequately presented in the paper. For example, the authors start from the concept of the Rwandan Environmental Management Authority (REMA) in 2006 [4], which states that Rwanda prefers to approach the green economy, with various environmental conservation policies and specified implementation plans [5, 6].

The research methodology is appropriate, respectively the authors of the research obtain results based on the established working hypotheses, by performing two tests for an inverted U-shaped model, as well as by the ARDL method for cointegration. These being research tools and methods appropriate to the topic of the paper.

The results presented by the research authors confirm “the short-term and long-term cointegration relations of the variables, where CO2-GDPc and CO2-AGRc are relatively decoupled and the causal relationship is significantly negative from GDPc and AGRc to CO2, respectively are confirmed. However, in order to clearly reflect the personal scientific contribution, we suggest to the authors of the research to complete the presentation of the research with the personal scientific results both in the literature and from the application point of view for the institutions in Rwanda, respectively for the economic sectors.

The conclusions are presented by the authors of the research, respectively as a result of the research the author’s state that the government and partners in the sector "should support agriculture towards a sustainable status, namely the consolidation of agroeconomics and climate change mitigation". However, as I mentioned in the presentation of the research results, we propose to mention in the paper personal scientific contributions to both the literature and the pragmatic utility for institutions in Rwanda.

Round 2

Reviewer 1 Report

All comments have been properly addressed.

This manuscript is a resubmission of an earlier submission. The following is a list of the peer review reports and author responses from that submission.

Round 1

Reviewer 1 Report

The topic presented in this work is really interesting. However several challenges are strictly required to make acceptable the manuscript.

Going through some major comments:

  1. Abstract has inappropriate structure. I suggest to answer the following aspects: - general context - novelty of the work - methodology used - main results. It should be more concise. Please look at guide for authors
  2. Introduction presents interesting information, but could be improved. Section 1 should be broader, and take the topic form general perspective.  Being agriculture the most relevant sector of the bioeconomy, I would suggest the authors to add at least a paragraph where introducing the need also for bioeconomy development.

Some references to start with:

Enabling a sustainable and prosperous future through science and innovation in the bioeconomy at Agriculture and Agri-Food Canada

Transitioning towards the bio-economy: Assessing the social dimension through a stakeholder lens

Transition to a bioeconomy: Perspectives from social sciences

Towards a sustainable forest-based bioeconomy in Italy: Findings from a SWOT analysis

Enabling a transformation to a bioeconomy in New Zealand

  1. Methodology is unclear. Initially a short resume can be proposed to explain several steps. It should be better linked to the existing literature. what is its potential? its limit?
  2. Results are not always linked to the methodology. Please define the relationship and relate your finding with the relevant literature.
  3. Conclusion are succinct. Please provide practical implications, limitations and future research needs.

Author Response

RESPONSE TO REVIEWER 1 COMMENTS

The topic presented in this work is really interesting. However several challenges are strictly required to make acceptable the manuscript.

 Going through some major comments:

Point 1 Abstract has inappropriate structure. I suggest answering the following aspects: - general context - novelty of the work - methodology used - main results. It should be more concise. Please look at guide for authors.

Response 1: We thank the reviewer for pointing out the issue of our paper’s abstract structure. Following the suggestion, we have revised the manuscript abstract according to the author’s guidance format provided by the journal. Now the general context, the novelty of the work, the methodology used, and the main results are cleared in our revised abstract of 197 words.

Point 2 Introduction presents interesting information, but could be improved. Section 1 should be broader, and take the topic form general perspective.  Being agriculture the most relevant sector of the bioeconomy, I would suggest the authors to add at least a paragraph where introducing the need also for bioeconomy development.

Some references to start with:

  • Enabling a sustainable and prosperous future through science and innovation in the bioeconomy at Agriculture and Agri-Food Canada
  • Transitioning towards the bio-economy: Assessing the social dimension through a stakeholder lens
  • Transition to a bioeconomy: Perspectives from social sciences
  • Towards a sustainable forest-based bioeconomy in Italy: Findings from a SWOT analysis
  • Enabling a transformation to a bioeconomy in New Zealand

Response 2 We really appreciate the reviewer's carefulness and the article he/she recommended to us. We have read the articles carefully and benefit a lot from the excellent and detailed work. The bioeconomy as the generation, utilization, and biological protection tool, knowledge, and novelty, to provide expertise and services across all economic area is integrated into our research, and as requested, other literature reviews including the suggested above also were added for the Section 1 broadening and topic general perspective.

Point 3 Methodology is unclear. Initially, a short resume can be proposed to explain several steps.

Response 3 We thank the reviewer for the constructive comment; an initial resume explaining several steps of our methodology on each approach used was integrated.

Point 4 Results are not always linked to the methodology. Please define the relationship and relate your findings with the relevant literature.

Response 4 we appreciate the reviewer's constructive concern on the linkage between our research results, methodology, and relevant literature.  We have accordingly revised the research results with a comprehensive liaison on the research methodology and comparisons with the relevant literature.

Point 5 The Conclusion is succinct. Please provide practical implications, limitations and future research needs.

Response 5 we thank the reviewer for pointing out the issue of the succinct conclusion of our research. Following the suggestion, we have revised the whole conclusion with the integration of practical implications, limitations, and future research needs.

Reviewer 2 Report

As I understand, the manuscript investigate the impact of capital accumulation on economic structures and, hence, on agricultural development [lines 54-57]. An empirical exercise is presented to investigate the correlation between the relevance of agriculture activities (agriculture revenues per capita), investments (gross capital formation per capita) and income (gdp per capita) [Table 1], and so to predict future revenues from agriculture. The empirical exercise focuses on Rwanda's data between 1961 to 2017 and presents forecasts until 2025. It is concluded that income, investments and agricultural development are mutually determined [lines 331-332]. The recommendation is therefore to design an extended mix of policy, that is, not limited to agriculture, to support agricultural development.

The manuscript considers a very relevant topic in economic development, especially for the less-industrialized countries, but unfortunately the presentation is unclear and insufficient to support the conclusions in my opinion. Here below are my comments on the improvements that I consider crucial before the manuscript can be considered for publication.

  1. Extensive rephrasing and language editing is necessary. It is really a hard work to read and comprehend the whole manuscript. A professional editing of language is strongly recommended.
  2. The Journal has a wide and various readership that does not necessarily have a strong background in time-series analysis. I would recommend to entirely restructure the manuscript so that it can be more palatable for a larger readership. Although a bit unusual, I would outline the paper according to the steps in the empirical strategy, mixing to report methods, data and results for each step.
  3. The background presented sounds poor. I recommend additional literature review, especially to provide a stronger theoretical support to the empirical exercise.
  4. Given the research question presented in the introductory section and the general mood of the conclusions, I was wondering whether it would be more appropriate to identify gdp per capita as the explained variable and the per-capita revenues from agriculture as one of the explanatory variables. This would demand, of course, for entirely restructuring the empirical exercise.
  5. The predictions presented in Table 11 are sometimes quite far from the real data observed in the past. I was wondering whether the structure of the empirical model is too simple to capture temporary processes and events. I would recommend to include at least a binary measure of exogenous and local shocks that can capture an unpredicted variability of data.

Author Response

RESPONSE TO REVIEWER 2 COMMENTS

As I understand, the manuscript investigate the impact of capital accumulation on economic structures and, hence, on agricultural development [lines 54-57]. An empirical exercise is presented to investigate the correlation between the relevance of agriculture activities (agriculture revenues per capita), investments (gross capital formation per capita) and income (gdp per capita) [Table 1], and so to predict future revenues from agriculture. The empirical exercise focuses on Rwanda's data between 1960 to 2017 and presents forecasts until 2025. It is concluded that income, investments and agricultural development are mutually determined [lines 331-332]. The recommendation is therefore to design an extended mix of policy, that is, not limited to agriculture, to support agricultural development.

The manuscript considers a very relevant topic in economic development, especially for the less-industrialized countries, but unfortunately the presentation is unclear and insufficient to support the conclusions in my opinion. Here below are my comments on the improvements that I consider crucial before the manuscript can be considered for publication.

Point 1 Extensive rephrasing and language editing is necessary. It is really a hard work to read and comprehend the whole manuscript. A professional editing of language is strongly recommended.

Response 1 We thank the reviewer for pointing out the issue of our paper language readability and recommendation of professional editing of language. Following the suggestion, we have revised the whole paper, and the language editing for the manuscript has been done to ensure that paper is more readable and concise.

Point 2 The Journal has a wide and various readerships that do not necessarily have a strong background in time-series analysis. I would recommend to entirely restructuring the manuscript so that it can be more palatable for a larger readership. Although a bit unusual, I would outline the paper according to the steps in the empirical strategy, mixing to report methods, data and results for each step.

Response 2 Many thanks for the reviewer for the constructive concern of the journal audience and paper outline. We did an entirely manuscript restructuring so that it can be more palatable for a broader readership. The paper outline also is improved as suggested, and each step our data results is followed by a comprehensive explanation.

Point 3 The background presented sounds poor. I recommend additional literature review, especially to provide a stronger theoretical support to the empirical exercise.

Response 3 We thank the reviewer for valuable suggestions. As requested, another literature review is added to enhance the research background.

Point 4 Given the research question presented in the introductory section and the general mood of the conclusions, I was wondering whether it would be more appropriate to identify gdp per capita as the explained variable and the per-capita revenues from agriculture as one of the explanatory variables. This would demand, of course, for entirely restructuring the empirical exercise.

Response 4 We appreciate the reviewer's suggestion. However, our research aims intended to: Identifying the relationship status (short term, long term or none) existence from Rwandan economy and investments to its agriculture sector; precise the relationship and causality directions of each explanatory variable to the current and future agriculture sector to classify the existing weaknesses, strengths, and opportunities; and propose the strategies for agriculture sector improvement. Thus changing the explained variable (agr) to become an explanatory variable goes beyond the scope of this paper, loose of our novelty, and we will no longer be able to identify the economic competitiveness of the Rwandan agriculture land-use sector to other economic activities land-use sectors neither identify the issues existing in the sector. Including the sector's workforce abdication, inadequate investments, elder population rate, and low-level skills in the sector, then propose some measures to overcome those issues.

Consider lines 14, 113, 475, 504 and 521.

Point 5 The predictions presented in Table 11 are sometimes quite far from the real data observed in the past. I was wondering whether the structure of the empirical model is too simple to capture temporary processes and events. I would recommend including at least a binary measure of exogenous and local shocks that can capture an unpredicted variability of data

Response 5 we thank the reviewer for the concern about some discrepancies the real data observed in the past and the predicted. We have include in our work an explanation of that discrepancy as follow and the prediction accuracy as follow: The predictions presented in Table 13 Agrc production forecast to 2025, present some discrepancies with the real data observed in the past; this was due to the uncontrolled chocks on the Rwandan agriculture trend as other economic and social areas. Resulted from the years 1972-1975, 1980-1990, 1990 to 1997 periods of internal conflicts, boosting rural area exodus for other economic activities in the large cities, and civil wars with the 1994 genocide, respectively [21], [22]. However, the MAPE, MAE, and RMSE interval errors values equal to 10.09%, 11.23, and 15.34, respectively, are acceptable and considered to produce good forecast efficiency according to the previous researches [46]

Reviewer 3 Report

About the submission with the title "Agriculture forecast relationship and causality with gross capital formation and economic growth in Rwanda" I have the following concerns:

This work could be an interesting contribution for several stakeholders, however in the present version I feel a little lost when I move from one section to another.

I suggest a clear explanation in the abstract (and after deeper in the introduction section) about the main motivations, objectives and methodologies (it is needed to clarify here the several steps made and to link them with the objectives) and the main insights (the text should be clear and specific, do not forget this is the first contact of the readers with the work). For example, for a first contact how the authors find this statement "Income is more inelastic than gross capital formation; thus, the per capita GDP is a more significant...."? And what means "...variables linkage is manifested. Therefore, an improvement of agriculture production requires...."?

I suggest, also, significant improvements in the literature review. A significant effort are needed, also, to update the literature.

The section 2 needs to be rewritten. The authors should begin with the objectives statement, after presenting and linking the methodologies adopted and finally benchmark the approaches chosen with other available (in fact, any econometric model can be used to make predictions. This could be made in the beginning of the section 2. After this, the authors should explain specifically each step supported with scientific literature already published.

This clarifications are, in fact, needed, because, for example, in the abstract the authors wrote "The NGBM-OP present reliable results of MAE...", but begin with with GM and ARIMA "...autoregressive integrated moving average (ARIMA) techniques are for the forecast process." in the section 2!

After in the line 102 wrote "If adding LGDPc2 results on insignificant adjusted R2, we conclude that the inclusion of LGDPc2 in...", but in the results appears a quadratic model!

In the all section 2 the authors wrote about results from statistics simulations (tests,...) but the results tables and figures only appear in the section 3. The, authors should explain clearly only the methodologies in the section 2 and presenting the all results in the section 3.

In the section 3, the authors should present the methodology (the model, equation, approach, ...) behind each table and figure and explain why each table and figure is important for the objectives of the research.

For example, in the present version I was unable to understand oii your ARIMA model is (0,0,0) or (1,1,1), or .....

The work has potential, but in the present version it is very hard to follow.

In the conclusions section the authors need to present the main implication for the several stakeholders and policy suggestions.

Author Response

RESPONSE TO REVIEWER 3 COMMENTS

About the submission with the title "Agriculture forecast relationship and causality with gross capital formation and economic growth in Rwanda" I have the following concerns:

This work could be an interesting contribution for several stakeholders, however in the present version I feel a little lost when I move from one section to another.

Point 1 I suggest a clear explanation in the abstract (and after deeper in the introduction section) about the main motivations, objectives and methodologies (it is needed to clarify here the several steps made and to link them with the objectives) and the main insights (the text should be clear and specific, do not forget this is the first contact of the readers with the work). For example, for a first contact how the authors find this statement "Income is more inelastic than gross capital formation; thus, the per capita GDP is a more significant...."? And what means "...variables linkage is manifested. Therefore, an improvement of agriculture production requires...."?

Response 1 we thank the reviewer for pointing out the issue of our paper’s abstract, objectives, and methodologies structures. Following the suggestion, we have revised the whole manuscript according to the author’s guidance format provided by the journal. Now the general context, the novelty of the work, the methodology used, and the main results are cleared in our revised work.

Point 2 I suggest, also, significant improvements in the literature review. A significant effort is needed, also, to update the literature.

Response 2 We thank the reviewer for valuable suggestions. As requested, a significant effort is added to update the literature, and another literature review was integrated to enhance the research background.

Point 3 The section 2 needs to be rewritten. The authors should begin with the objectives statement, after presenting and linking the methodologies adopted and finally benchmark the approaches chosen with other available (in fact, any econometric model can be used to make predictions. This could be made in the beginning of the section 2. After this, the authors should explain specifically each step supported with scientific literature already published.

Response 3 Thanks for your comments and constructive suggestion. All of the questions have been addressed in the revised version linking the adopted methodologies, and each step is explained and supported with scientific literature already published.

Point 4 This clarifications are, in fact, needed, because, for example, in the abstract the authors wrote "The NGBM-OP present reliable results of MAE...", but begin with with GM and ARIMA "...autoregressive integrated moving average (ARIMA) techniques are for the forecast process." in the section 2!

Response 4 We thank the reviewer for valuable concern. As requested, more clarifications on those model performances were more explained in the revised version and show why we decide to opt for  NGBM-OP rather than ARIMA(3,1,2) and GM(1,1).

Consider lines 228-281, 410, 412, and 414.

Point 5 After in the line 102 wrote "If adding LGDPc2 results on insignificant adjusted R2, we conclude that the inclusion of LGDPc2 in...", but in the results appears a quadratic model!

Response 5 Thanks for your concern about the model test significance. Really this was a typing error as all tests for quadratic model are significance.

Consider lines: 195, 373-377, and Figure 4

Point 6 In the all section 2 the authors wrote about results from statistics simulations (tests,...) but the results tables and figures only appear in the section 3. The, authors should explain clearly only the methodologies in the section 2 and presenting the all results in the section 3.

Response 6 Thanks for your concern. In the revised version, the methodology is clearly explained in section 2 and presenting all results in section 3.

Consider lines:  137-281 section 3 (Materials and Methods) and 282 section 3 (3. Results)

Point 7 In the section 3, the authors should present the methodology (the model, equation, approach ...) behind each table and figure and explain why each table and figure is important for the objectives of the research. For example, in the present version I was unable to understand oii your ARIMA model is (0, 0, 0) or (1, 1, 1), or

Response 7 We thank the reviewer for valuable concern. In the new version behind each table and figure, explanations are provided on each table and figure explaining it is essential for the research objectives. And the appropriate ARIMA for our series to be compared with NGBM-OP is ARIMA (3, 1, 2)

Point 8 The work has potential, but in the present version it is very hard to follow. In the conclusions section the authors need to present the main implication for the several stakeholders and policy suggestions.

Response 7 We thank the reviewer for pointing out the issue of our paper readability and recommendation. Following the suggestion, we have revised the whole paper, according to all requirements, and the new version is improved.

Thank you for your valuable time to review our manuscript.

Round 2

Reviewer 1 Report

Dear Authors,

thank you for your revised version. The manuscript is much improved.

A little effort, however, is still required to make it publishable. 

Line 520. Your implications are in line with the following article: 

Falcone, P. M., & Imbert, E. (2019). Tackling Uncertainty in the Bio-Based Economy. International Journal of Standardization Research (IJSR), 17(1), 74-84.

Author Response

RESPONSE TO REVIEWER 1 COMMENTS

Thank you for your revised version. The manuscript is much improved. A little effort, however, is still required to make it publishable.

Point 1 Line 520: Your implications are in line with the following article: Falcone, P. M., & Imbert, E. (2019), Tackling Uncertainty in the Bio-Based Economy. International Journal of Standardization Research (IJSR), 17(1), 74-84.

Response 1: We thank the reviewer for pointing out this issue in our paper’s policy implications part. Now the paper’s policy implications are revised point by point. In the new version, they are more focused on research model results, the main purpose of the research, as well as several official statistical reports consulted.

Reviewer 2 Report

I thanks the authors for their effort in meeting my comments and motivate their point where my comments were diverging from their research settings. The last version is much easier to follow than the previous one, and some issues in the evidence are appropriately discussed. Nonetheless, an additional effort in editing would be appreciated. Furthermore, I would recommend again to extend the literature review, especially in the introductory and the concluding sections, so that the piece of research presented in this manuscript can be better placed in the literature and the results can be better compared to the existing evidence and common guidelines to agriculture development. 

Author Response

I thank the authors for their effort in meeting my comments and motivate their point where my comments were diverging from their research settings. The last version is much easier to follow than the previous one, and some issues in the evidence are appropriately discussed. Nonetheless, an additional effort in editing would be appreciated.

Point 1 Furthermore, I would recommend again extending the literature review, especially in the introductory and the concluding sections, so that the piece of research presented in this manuscript can be better placed in the literature and the results can be better compared to the existing evidence and common guidelines to agriculture development.

Response 1 Many thanks for the reviewer for the constructive concern of extending the literature review. Following the suggestion, we have revised the introductory and conclusion parts by integrating other agro-economic previous researches published and official statistical reports to extending the literature review. After that, our results in the conclusion part are compared to the existing evidence.

Reviewer 3 Report

The authors made a great effort.

I suggest the authors improve the literature review.

Author Response

The authors made a great effort.

Point 1 I suggest the authors improve the literature review

Response 1 Many thanks for the reviewer for the constructive concern of the literature review improvement. Following the suggestion in the new version, we have an improved literature review part by integrating other agro-economic previous researches published papers, and official statistical reports, after that our results in the conclusion parts are compared to the existing evidence.
